# Surgeon- and hospital-level variation in wait times for scheduled non-urgent surgery in Ontario, Canada: A cross-sectional population-based study

**Pieter de Jager**[1]ᴼ*, **Dionne Aleman**[2‡], **Nancy Baxter**[3‡], **Chaim Bell**[4,5‡], **Merve Bodur**[2‡], **Andrew Calzavara**[6ᴼ], **Robert Campbell**[7‡], **Michael Carter**[2‡], **Scott Emerson**[8‡], **Anna Gagliardi**[9‡], **Jonathan Irish**[10‡], **Danielle Martin**[11‡], **Samantha Lee**[6‡], **Marcy SaxeBraithwaite**[12‡], **Pardis Seyedi**[2‡], **Julie Takata**[13‡], **Suting Yang**[14‡], **Claudia Zanchetta**[15‡], **David R. Urbach**[13ᴼ]

1 Department of Anesthesia, Pain Management & Peri operative Medicine, Dalhousie University, Halifax, NS, Canada, 2 Department of Mechanical and Industrial Engineering, University of Toronto, Toronto, Ontario, Canada, 3 University of Melbourne School of Population and Global Health, Carlton, Australia, 4 Department of Medicine, Sinai Health System, Toronto, Ontario, Canada, 5 University of Toronto Temerty Faculty of Medicine, Department of Medicine, Toronto, Ontario, Canada, 6 ICES Central, Toronto, Ontario, Canada, 7 Department of Ophthalmology, Queen's University, Kingston, Ontario, Canada, 8 Human Early Learning Partnership, School of Population and Public Health, The University of British Columbia, Vancouver, BC, Canada, 9 Toronto General Research Institute, Toronto, Ontario, Canada, 10 Department of Otolaryngology, Head and Neck Surgery, University Health Network, Toronto, Ontario, Canada, 11 Department of Family and Community Medicine, University of Toronto, Toronto, Ontario, Canada, 12 Nova Scotia Health Authority, Halifax, NS, Canada, 13 Department of Surgery, Women's College Hospital, Surgery, Toronto, Ontario, Canada, 14 Ontario Health, Toronto, Ontario, Canada, 15 Access to Care, Ontario Health (Cancer Care Ontario), Toronto, Ontario, Canada

ᴼ These authors contributed equally to this work.
‡ These authors also contributed equally to this work.
* Pieter.deJager@nshealth.ca

**Data Availability Statement:** The raw dataset from this study is held securely in coded form at ICES. While legal data sharing agreements between ICES

## Abstract

### Background

Canadian health systems fare poorly in providing timely access to elective surgical care, which is crucial for quality, trust, and satisfaction.

### Methods

We conducted a cross-sectional analysis of surgical wait times for adults receiving non-urgent cataract surgery, knee arthroplasty, hip arthroplasty, gallbladder surgery, and non-cancer uterine surgery in Ontario, Canada, between 2013 and 2019. We obtained data from the Wait Times Information System (WTIS) database. Inter- and intra-hospital and surgeon variations in wait time were described graphically with caterpillar plots. We used non-nested 3-level hierarchical random effects models to estimate variation partition coefficients, quantifying the proportion of wait time variance attributable to surgeons and hospitals.

and data providers (e.g., healthcare organizations and government) prohibit ICES from making the dataset publicly available, access may be granted to those who meet pre-specified criteria for confidential access, available at www.ices.on.ca/DAS (email: das@ices.on.ca). The full dataset creation plan and underlying analytic code are available from www.ices.on.ca/DAS (email: das@ices.on.ca) upon request, understanding that the computer programs may rely upon coding templates or macros that are unique to ICES and are therefore either inaccessible or may require modification.

**Funding:** This project was funded by the Canadian Institutes of Health Research through a project grant (no. PJT-166108). The funders had no role in study design, data collection and analysis, decision to publish, or preparation of the manuscript. This study was supported by ICES, which is funded by an annual grant from the Ontario Ministry of Health (MOH) and the Ministry of Long-Term Care (MLTC).

**Competing interests:** The authors have declared that no competing interests exist.

## Results

A total of 942,605 procedures at 107 healthcare facilities, conducted by 1,834 surgeons, were included in the analysis. We observed significant intra- and inter-provider variations in wait times across all five surgical procedures. Inter-facility median wait time varied between six-fold for gallbladder surgery and 15-fold for knee arthroplasty. Inter-surgeon variation was more pronounced, ranging from a 17-fold median wait time difference for cataract surgery to a 216-fold difference for non-cancer uterine surgery. The proportion of variation in wait times attributable to facilities ranged from 6.2% for gallbladder surgery to 23.0% for cataract surgery. In comparison, surgeon-related variation ranged from 16.0% for non-cancer uterine surgery to 28.0% for cataract surgery.

## Implications

There is extreme variability in surgical wait times for five common, high-volume, non-urgent surgical procedures. Strategies to address surgical wait times must address the variation between service providers through better coordination of supply and demand. Approaches such as single-entry models could improve surgical system performance.

## Background

Timely access to surgical care has gained increasing policy and academic scrutiny [1–3]. Access to timely care impacts patient outcomes, trust in the healthcare system and user satisfaction [4]. Canada has consistently ranked well below its economic peers in access to care [5].

Wait time targets and reliable data for wait times have recently become available in multiple Canadian jurisdictions [6]. Policy and scholarly attention have focused on whether people exceed benchmarked wait times, with scarce inquiry into the variation of wait time between providers. Therefore, it is unclear whether the main problem is that wait times are too long in general or whether there is simply excessive variation in wait times, with some patients waiting longer than recommended targets and others receiving care quickly.

To answer this question, we measured variation in surgical wait times for commonly performed elective surgical procedures between hospitals and surgeons in Ontario, Canada.

## Methods

### Study setting

The provincial government in Ontario (estimated population of 15.8 million as of 2023) pays hospital and physician fees for all medically necessary services [7,8]. Patients in Ontario may be referred to any surgeon at any hospital, usually through direct primary care physician-to-surgeon referral.

Ontario categorizes surgical procedures according to priority level, from 1 (emergency) to 4 (least urgent). Further, Ontario collects physician-reported data on two wait times: Wait 1 is defined as the time between the referral to the date of surgical consultation, and Wait 2 is defined as the time between the clinical decision to proceed with surgical treatment and the surgery date, subtracting any patient-related delays such as undergoing other treatments, change in medical status, or patient choice [9]. Ontario's benchmark for Wait 2 for the least

urgent category of most non-cancer conditions is within 184 days of deciding to proceed with surgery and remained constant throughout the study period [10].

## Data sources

We identified surgical procedures in the Wait Times Information System (WTIS) database, administered by Ontario Health—an agency created by the Government of Ontario to connect and coordinate the province's health care system. The database provides standardized and validated wait time tracking for surgeries in Ontario [8,11] and has been utilized in health systems research [12,13]. The WTIS provides data on 59 surgical wait time elements from over 3,300 surgeons across 132 healthcare sites in Ontario. For the procedures we studied, including cataract surgery, this covers the vast majority of these procedures in Ontario.

## Outcome and exposure

The primary outcome in this analysis was surgical wait time from the decision to operate until the date of surgery (Wait 2) for persons 18 years or older undergoing non-urgent (priority level 4) cataract surgery, knee arthroplasty, hip arthroplasty, gallbladder surgery, and non-cancer uterine surgery in Ontario between April 2013 and December 2019. The benchmark for all these procedures of 184 days for Wait 2 did not change during the study period. The period was selected to provide a representative period before the onset of the COVID-19 pandemic, reflecting a steady state not affected by systemic disruptions during the pandemic. We chose these procedures because they are among Ontario's most common surgical procedures and represent a spectrum of different types of care settings, medical specialties, and patient populations [13].

We excluded procedures occurring within one year of a previous procedure of the same type. Surgery on potentially bilateral sites, such as arthroplasty, could be intentionally staged, skewing the subsequent procedures' wait times and undermining the assumption of independence. Further, we excluded higher acuity (priority level 1 to 3) procedures or those missing data on priority level, non-cancer uterine surgery with male sex, procedures at low volume hospitals (<10 procedures) or low volume surgeons (< 10 procedures) during the study period, and procedures with outlier wait times (0 days or > 99$^{th}$ percentile). Table 1 provides a summary of the data management and exclusions. Considering the non-urgent nature of the procedures, it is likely that the few excluded records with zero wait time either contained inaccurately recorded dates or reflected deviations from the usual care pathways. Further, we excluded these observations to avoid undermining modelling assumptions.

This study focuses on an analysis of Wait 2 times, for which there were no missing data in WTIS. Since Wait 2 times are defined as the time from the decision to proceed with surgery and the date of surgery, this interval comprises a relatively homogenous surgical population awaiting surgery and excludes downstream factors which impact wait time, such as additional patient work-up, investigations and diagnostics. Analyses of Wait 1 times yielded similar results and are not reported here.

Our primary exposures of interest for the analysis were the individual surgeons performing the surgical procedures and the healthcare facilities where the procedures occurred. Most surgeons in Ontario work as independent practitioners and manage their own "wait list." Although not all procedures were performed in hospitals, we used the term "hospital" to designate the location of each procedure, and we used the term "surgeon" to describe the physician who performed the procedure.

**Table 1. Summary of data management and exclusions.**

| | TOTAL | Cataract | Knee arthroplasty | Hip arthroplasty | Gallbladder surgery | Non-cancer uterine |
|---|---|---|---|---|---|---|
| | N = 1,551,339 | N = 897,874 | N = 179,135 | N = 101,476 | N = 128,418 | N = 244,436 |
| Procedure occurring within one year of previous procedure of the same type | 371,961 (24.0%) | 339,310 (37.8%) | 16,360 (9.1%) | 6,158 (6.1%) | 371 (0.3%) | 9,762 (4.0%) |
| Priority level 1–3 or missing | 214,898 (13.9%) | 42,372 (4.7%) | 34,861 (19.5%) | 22,816 (22.5%) | 50,510 (39.3%) | 64,339 (26.3%) |
| Non-cancer uterine procedures with male sex indicated | 416 (0.0%) | 0 (0.0%) | 0 (0.0%) | 0 (0.0%) | 0 (0.0%) | 416 (0.2%) |
| Zero wait time | 10,555 (0.7%) | 7,050 (0.8%) | 1,528 (0.9%) | 818 (0.8%) | 713 (0.6%) | 446 (0.2%) |
| Wait time > 99th percentile (of remainder) | 9,616 (0.6%) | 5,153 (0.6%) | 1,277 (0.7%) | 723 (0.7%) | 773 (0.6%) | 1,690 (0.7%) |
| Procedure at low volume hospital or by low-volume surgeon (< 10 procedures) | 1,288 (0.1%) | 87 (0.0%) | 101 (0.1%) | 147 (0.1%) | 498 (0.4%) | 455 (0.2%) |
| INCLUDED | 942,605 (60.8%) | 503,902 (56.1%) | 125,008 (69.8%) | 70,814 (69.8%) | 75,553 (58.8%) | 167,328 (68.5%) |

Notes: Wait Time Information System (WTIS) classifies procedure priority level. Level 1 is the most urgent, requiring immediate transfer to the operating room. This is followed by levels 2, 3, and 4, which require urgent, semi-urgent, and elective surgical intervention. Non-cancer uterine surgery consists of a mixture of hysterectomy (28%), hysteroscopic endometrial ablation (16%) and other benign diseases (56%). 'Other benign diseases' comprise resection of endometrial polyps, myomectomy, and adhesion lysis.

Gallbladder surgery constitutes primarily cholecystectomy with a small number of cholecystostomy and choledochotomy.

## Statistical analysis

We reported descriptive statistics for each procedure, including the number of hospitals, surgeons, and procedures performed. For each surgical procedure, we also summarized wait times, patient age, sex, administrative region in Ontario (Toronto, Greater Toronto and Hamilton Area, Eastern, Western, and Northern), and hospital teaching status.

While variation in wait times is due to many factors, we were particularly interested in quantifying the variation attributable to the surgeon or hospital that performed the surgery. To visualize the variation in surgical wait times between individual surgeons and hospitals for each procedure, we first created caterpillar plots (box plots ordered by the mean value) that presented the mean, median value, and 10th, 25th,75th, and 90th percentiles of wait time.

For statistical modelling, we assessed the distribution of the conditional residuals under various transformations of the outcome. A squared logarithmic transformation with the most extreme 1% of wait times excluded resulted in an approximately normal distribution, with residual plots showing no sign of heteroscedasticity. We fitted non-nested (cross-classified) 3-level hierarchical random effects models, with the squared logarithm of the wait time as the outcome, and one random intercept for the hospital and one random intercept for the surgeon. We used a non-nested structure to accommodate physicians who operated at more than one hospital, either concurrently or by switching hospitals during the study period. We treated multi-site hospitals or hospitals that merged or changed their identifying number during the study period as a single hospital. Since the analysis aimed to quantify the observed variation in wait time due to surgeons and hospitals, we did not include explanatory fixed effects in the models. Likelihood ratio tests comparing the two-intercept model to models having one of each intercept, and each one-intercept model to a null model with no random effects, were used to determine if variances were significantly greater than zero. We used the models to analyze variance components by estimating the Variance Partition Coefficient (VPC) for the surgeon and the hospital effects [14]. Similar approaches have been used elsewhere [15].

The VPC represents the proportion of the total observed individual variation in the outcome attributable to between-cluster variation and, in the presented models, is equivalent to

an intraclass correlation coefficient (ICC), i.e., the correlation in the outcome between two events randomly selected from the same cluster. For example, the VPC for the surgeon estimates is the proportion of variation in the wait time outcome attributable to the fact that a particular surgeon operated and not due to the hospital, patient-related factors, or any other causes of variation. If every surgeon had the same wait time distribution, the VPC for the surgeon estimate would be 0. If all variation were due solely to different surgeons having different wait times independent of any other factors (such that every procedure performed by a given surgeon had the same wait time), the VPC of the surgeon estimate would be 1. The residual variance is the variance in observed wait time attributable to all factors other than hospital or surgeon cluster-level factors, such as patient-level factors (like comorbid medical conditions or socioeconomic status, although all procedures were classified as the least urgent priority level), regional differences, patient preference, and random variation. We used a significance level of 0.05 to denote statistical significance. Analyses were performed using SAS 9.4. The Reporting of Studies Conducted Using Observational Routinely-collected Data guidelines were followed in presenting our analysis [16].

## Ethics approval

This study was approved by the Women's College Hospital and Sunnybrook Health Sciences Centre Research Ethics Boards. Data were accessed on 26 September 2023. The authors did not have access to information that could identify individual patients. The data were fully anonymized, linked using unique encoded identifiers, and analyzed at ICES. ICES is an independent, non-profit research institute whose legal status under Ontario's health information privacy law allows it to collect and analyze health care and demographic data without consent for health system evaluation and improvement. The use of the data in this project is authorized under section 45 of Ontario's Personal Health Information Protection Act (PHIPA) and does not require review by a Research Ethics Board.

## Results

In total, there were 107 hospitals and surgical centres, as well as 1,834 surgeons. Of the 942,605 procedures included in the analysis, the majority were cataract surgery (53.5%), followed by non-cancer uterine surgery (17.8%), knee arthroplasty (13.3%), gallbladder surgery (15.0%) and hip arthroplasty (14.1%).

Table 2 presents the number of hospitals and surgeons performing each procedure and surgical volumes. Variation in surgical wait time varied markedly for all five procedures under consideration, from 1–235 days for gallbladder surgery to 1–548 days for knee arthroplasty. Patients having gallbladder and non-cancer uterine surgery were younger on average. The proportion of procedures conducted in teaching hospitals ranged from 19.5% for gallbladder surgery to 37.7% for hip arthroplasty.

Fig 1 illustrates the variation in wait time within and between each hospital by procedure, ranked from lowest to highest average wait time. Across all five procedures, there was clear within- and between-hospital variation in surgical wait time. For cataract surgery, median surgical wait time varied between hospitals from 27 days to 188 days, representing a seven-fold difference in inter-hospital wait time. Similarly, considerable variation was seen for knee arthroplasty (24 to 368 days, 15-fold difference), hip arthroplasty (43 to 298 days; 7-fold difference); gallbladder surgery (19 to 120 days; 6-fold difference) and non-cancer uterine surgery (11 to 117 days; 11-fold difference).

Inter- and intra-surgeon wait time variability was even more stark (Fig 2). Surgeon median wait time for cataract surgery ranged from 15 to 249 days, a 17-fold difference. Inter-surgeon

**Table 2. Characteristics of patients, surgeons, and hospitals by procedure.**

| | | Cataract | Knee arthroplasty | Hip arthroplasty | Gall bladder surgery | Non-cancer uterine |
|---|---|---|---|---|---|---|
| | | N = 503,902 | N = 125,008 | N = 70,814 | N = 75,553 | N = 167,328 |
| **Hospitals** | Number of hospitals | 68 | 67 | 65 | 98 | 86 |
| | Mean number of procedures per hospital | 7 410 | 1 866 | 1 089 | 771 | 1 946 |
| | Median (25th-75th percentile) number of procedures per hospital | 4,948 (1,906–9,132) | 1,641 (653–2,523) | 937 (419–1,337) | 582 (200–1,054) | 1,495 (562–2,906) |
| | Mean number of surgeons per hospital | 7 | 6 | 6 | 7 | 10 |
| | Median (IQR25th-75th percentile) number of surgeons per hospital | 4 (2–9) | 6 (4–8) | 5 (4–7) | 6 (4–9) | 9 (4–15) |
| **Surgeons** | Number of surgeons | 316 | 340 | 305 | 512 | 663 |
| Procedures | Mean number of procedures per surgeon | 1 595 | 368 | 232 | 148 | 252 |
| | Median (25th-75th percentile) number of procedures per surgeon | 1,398 (526–2,346) | 300 (119–562) | 189 (80–327) | 101 (35–224) | 177 (74–351) |
| Wait time (days) | Mean (standard deviation) | 95.76 (77.0) | 118.52 (97.6) | 108.47 (86.8) | 49.91 (40.1) | 67.18 (50.5) |
| | Median (25th-75th percentile) | 75 (38–132) | 95 (47–162) | 87 (45–149) | 39 (21–66) | 55 (30–91) |
| | Range (Minimum–Maximum) | 1–399 | 1–548 | 1–473 | 1–235 | 1–286 |
| Patient age | Mean (standard deviation) | 71.65 (9.6) | 67.70 (9.2) | 66.93 (11.2) | 50.62 (15.8) | 46.27 (11.8) |
| | Median (25th-75th percentile) | 72 (66–78) | 68 (61–74) | 67 (60–75) | 51 (38–62) | 45 (38–53) |
| Patient sex | Female–number (%) | 282,539 (56.1%) | 77,013 (61.6%) | 38,308 (54.1%) | 54,684 (72.4%) | 167,328 (100.0%) |
| | Male–number (%) | 221,363 (43.9%) | 47,995 (38.4%) | 32,506 (45.9%) | 20,869 (27.6%) | 0 (0.0%) |
| Postal region of patients' residence | Eastern–number (%) | 95,460 (18.9%) | 25,002 (20.0%) | 15,307 (21.6%) | 11,319 (15.0%) | 21,995 (13.1%) |
| | GTHA–number (%) | 170,717 (33.9%) | 43,285 (34.6%) | 23,790 (33.6%) | 28,881 (38.2%) | 64,379 (38.5%) |
| | Toronto–number (%) | 87,590 (17.4%) | 16,027 (12.8%) | 9,645 (13.6%) | 10,877 (14.4%) | 28,466 (17.0%) |
| | Western–number (%) | 108,239 (21.5%) | 29,675 (23.7%) | 16,358 (23.1%) | 18,195 (24.1%) | 40,369 (24.1%) |
| | Northern–number (%) | 41,896 (8.3%) | 11,019 (8.8%) | 5,714 (8.1%) | 6,281 (8.3%) | 12,119 (7.2%) |
| Teaching hospital | number (%) | 121,968 (24.2%) | 36,939 (29.5%) | 26,692 (37.7%) | 14,745 (19.5%) | 45,683 (27.3%) |

Notes: Non-cancer uterine surgery consists of a mixture of hysterectomy (28%), hysteroscopic endometrial ablation (16%) and other benign diseases (56%). 'Other benign diseases' comprise resection of endometrial polyps, myomectomy, and adhesion lysis. Gallbladder surgery constitutes primarily cholecystectomy with a small number of cholecystostomy and choledochotomy. GTHA = Greater Toronto Hamilton Area. The term "hospital" is used to designate the location of each procedure, and we used the term "surgeon" to describe the physician who performed the procedure.

wait time variance was even higher for hip arthroplasty (16 to 382 days; 20-fold difference), knee arthroplasty (16 to 382 days; 24-fold difference), gallbladder surgery (1 to 133 days; 133-fold difference) and non-cancer uterine surgery (1–216 days; 216-fold difference).

Table 3 summarizes the results of the variance components analysis by procedure. The proportion of variation in wait time attributable to hospitals ranged from 6.2% for gallbladder surgery to 23.0% for cataract surgery. Wait time variation attributable to surgeons ranged from 16.0% for non-cancer uterine surgery to 28.0% for cataract surgery. Likelihood ratio tests determined that all surgeon- and hospital-level variances were significantly greater than 0 (P < 0.0001).

## Discussion

There is enormous variation in surgical wait time across both hospitals and surgeons for cataract surgery, knee arthroplasty, hip arthroplasty, gallbladder, and non-cancer uterine surgery in Ontario. This finding is particularly striking because all procedures we analyzed were the least urgent priority category for scheduled surgery used in Ontario and excluded potential

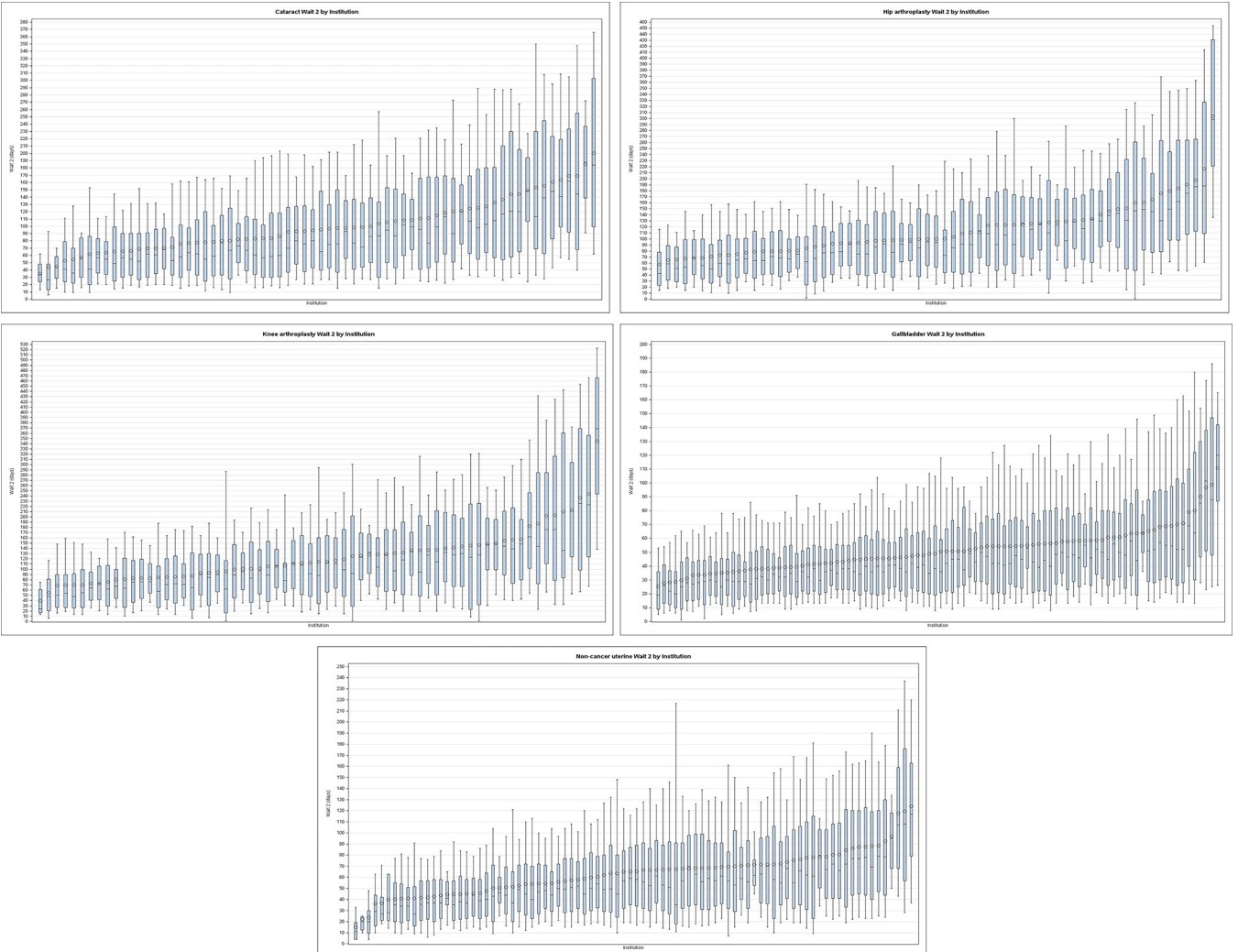

**Fig 1. Caterpillar plot of wait time by hospital, according to procedure.** Wait 2 time, ordered by average wait time (diamond) in days from shortest to longest, T-bars indicate 10[th] and 90[th] percentiles, shaded box indicates 25th to 75th percentile, horizontal line indicates median.

heterogeneity due to patient urgency or upstream factors that could impact wait time before the decision to proceed with surgery. This degree of variation cannot be explained based on clinical considerations.

Our analyses demonstrate that there is no such thing as a "wait list" for elective surgery in Ontario; rather, every patient awaiting elective surgery is effectively in a unique queue, separate from others in the system, with each queue having a wide range of wait-time variations. A meaningful amount of the variation in wait times is attributable to hospitals, ranging from 6.2% for gallbladder surgery to 23.0% for cataract surgery. Surgeon-related variation ranged from 16.0% for non-cancer uterine surgery to 28.0% for cataract surgery.

Wait times for surgery are commonly attributed to one or more of three factors–supply of services, demand for services, and coordination between supply and demand [2]. While a significant amount of public debate on improving surgical wait times in Canada has focused on increasing the supply of services through additional investment in existing not-for-profit hospitals or expansion of private for-profit surgical facilities [17]. Our findings are hypothesis-

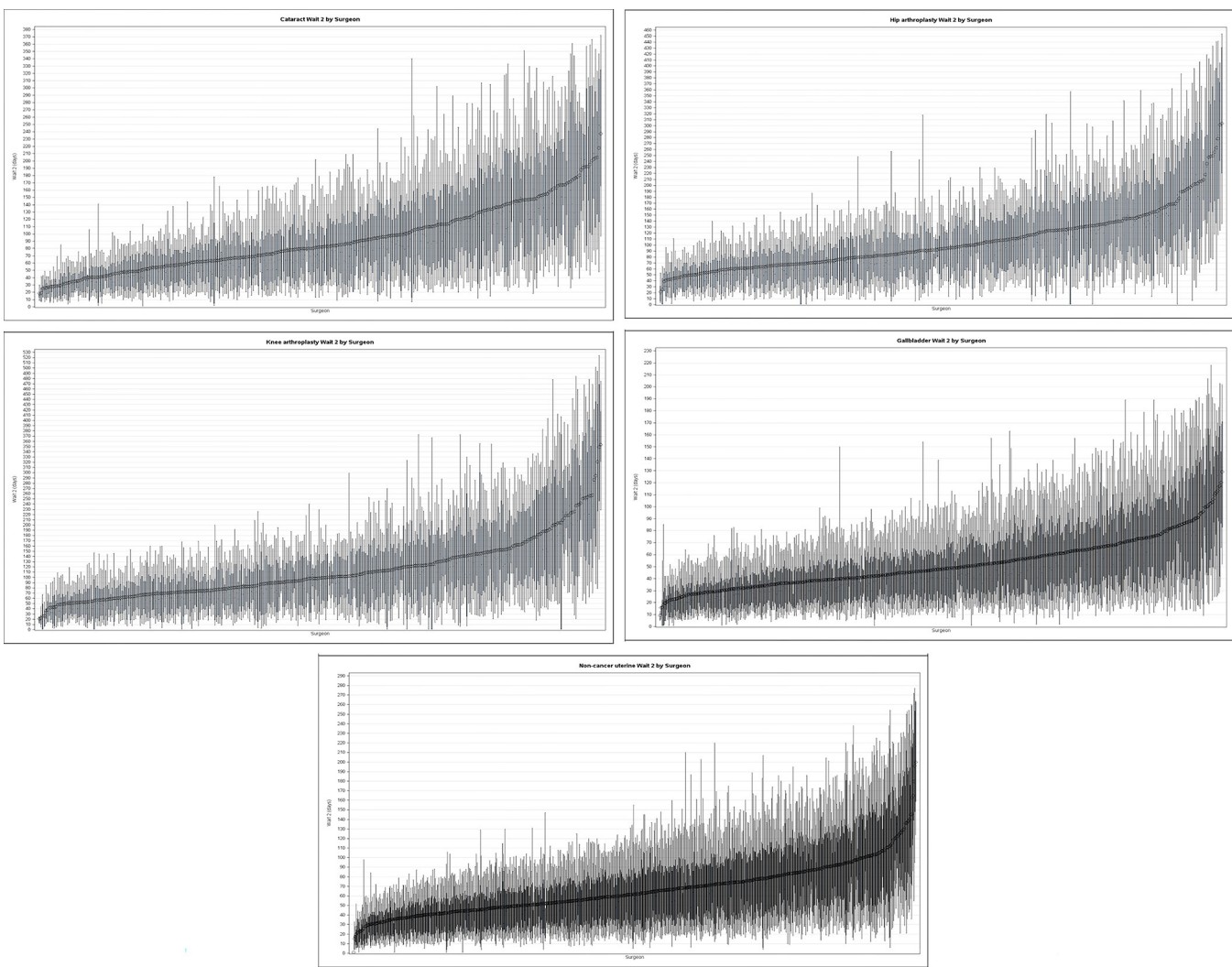

**Fig 2. Caterpillar plot of wait time by surgeon, according to procedure.** Wait 2 time, ordered by average wait time (diamond) in days from shortest to longest, T-bars indicate 10[th] and 90[th] percentiles, shaded box indicates 25th to 75th percentile, horizontal line indicates median.

generating, indicating that there might be efficiency gains from improving coordination between the supply and demand of services.

Our analysis suggests that the current system of accessing surgical care for these five high-volume elective procedures is allocatively inefficient. Allocative efficiency refers to a distribution of resources whereby resource allocation meets the consumer's needs [18]. Conceptually, allocative efficiency for elective surgery would be represented by a lack of significant variation of wait time between equivalent care providers: the probability of receiving surgical care within a given time frame would be uncorrelated with the provider, the caterpillar plots would demonstrate relatively uniform ordered box plots, and the provider-level VPCs would not be significantly greater than zero.

While some variation in wait time is inevitable, the considerable observed variation in wait time indicate additional drivers of wait time variation beyond patient need or differences in provider expertise. Earlier analyses did not demonstrate consistent evidence linking increased surgical wait time with social deprivation or morbidity [13]. Therefore, additional research is

**Table 3. Variance components analysis of hospital and surgeon random effects by procedure.**

| Procedure | Component of Variation | Variance | Variance partition coefficient (VPC) |
|---|---|---|---|
| Cataract | Surgeon | 21.68 | 0.28 |
| | Hospital | 17.82 | 0.23 |
| | Residual* | 37.83 | 0.49 |
| Knee arthroplasty | Surgeon | 12.96 | 0.18 |
| | Hospital | 7.33 | 0.10 |
| | Residual* | 49.87 | 0.71 |
| Hip arthroplasty | Surgeon | 11.69 | 0.19 |
| | Hospital | 5.54 | 0.09 |
| | Residual* | 44.58 | 0.72 |
| Gallbladder removal | Surgeon | 6.15 | 0.17 |
| | Hospital | 2.27 | 0.06 |
| | Residual* | 28.38 | 0.77 |
| Non-cancer uterine | Surgeon | 7.09 | 0.16 |
| | Hospital | 6.60 | 0.15 |
| | Residual* | 30.62 | 0.69 |

The outcome for all models was the squared logarithm of the Wait 2 time.

*Residual variance includes all remaining sources of variation after considering the specific surgeon and hospital performing the surgery.

required to elucidate the drivers of the observed variation, with a particular focus on the healthcare system, regional differences, and referral pathway design.

Finally, given the wide variation in the surgical wait times of patients treated within the same hospital (and even by the same surgeon), the reporting of mean or median wait times—or the proportion of patients not meeting waiting time targets—may be an inadequate measure of overall system performance in access to care, since it obscures the true extent of wait time variation. To the extent that confidence in Canadian health systems is threatened by a public perception that wait times for surgery are too long, it is vital to understand how those perceptions are formed. Public perceptions of access to surgical services in a health system are likely influenced most by those patients waiting the longest since these cases attract attention from the media, policy institutes, and the courts [19]. The extreme right tails in the caterpillar plots of wait times represent these patients. Average wait times would be meaningful if surgeon and hospital clinical and referral pathways were integrated systematically and patients awaiting surgery were part of a local or regional queue across pooled providers. There is a belief that long wait times reflect "shortages" of surgeons, operating room time, and hospital capacity [20]. However, given the significant variation in wait time between surgeons and hospitals, only increasing the supply of surgical services will not necessarily reduce the longest wait times without a coordinated system that better matches demand to supply.

A policy intervention promoted by many analysts and health system leaders that could reduce the extreme variation in wait times is the broader use of single-entry models and team-based care [21–23]. Single-entry models, also referred to as "central intake" models, are informed by queuing theory and can reduce excess variation in wait times for consultations, and also for surgery, by balancing clinical loads among surgeons and hospitals and smoothing out the mismatch between supply and demand. Single-entry referral models may improve system performance by reducing variation in wait time, improving allocative efficiency, and reducing inequity in access to timely surgical care. The principle underlying single entry models can also be applied to patients who have already been assessed by a surgeon and are

awaiting their surgical procedure. Assignment to surgery by the next available surgeon within a hospital or geographic region in a "team-based" or "shared care" model, as compared with restricting patients to have surgery by the same surgeons who provided the initial consultation, can further reduce variations in Wait 2 times. Without single-entry referral models, making policy decisions about increasing the supply of services is difficult since patients awaiting surgery are not participants in a single queue. From a surgeon's perspective, a single-entry system could reduce uncertainty in the demand for their service by ensuring a more reliable and steady flow of patients. Finally, it could also mitigate uncertainty in resource planning at a facility level.

Our study has several limitations. We utilized an administrative dataset to investigate variability in wait time. Comparisons with other administrative data found nearly 98% of procedure dates in WTIS matched an OHIP procedure claim, and the "decision to treat" date corresponded to a surgeon consult claim for over 80% of procedures analyzed [13]. Second, our analysis was limited to Ontario and scheduled non-urgent surgeries before the COVID-19 pandemic. However, Ontario accounts for approximately one-third of the Canadian population. Cases with greater clinical urgency, such as cancer surgery, may also limit generalizability, as care pathways, case mix, and clinical complexity can impact wait time dynamics differently. Similarly, our analysis did not account for highly specialized practices, such as robot-assisted surgeries, with limited supply due to skill shortages and the need for significant capital investment. Third, we excluded procedures with wait times at the extremes to allow transformation to a normal distribution; this might have biased our analysis towards underestimating the extent of variation. Fourth, although our variance components analysis showed that a significant proportion of wait time variation is attributable to surgeon and hospital cluster-level factors, most of the variation–as expressed in the residual variance–is secondary to other factors that our current analysis did not analyze. The protocol for this study was defined a priori to exclude fixed-effect covariates such as age, sex, marginalization, as well as surgical volume. Our intent was not to explain any of the reasons behind the observed variation in wait times. Rather, our purpose was to describe the extent of this variation, with the intent of stimulating the development and implementation of health policy interventions that can attenuate this effect. The explanations for the variations in wait times we observed are important and should be the subject of future research. Fifth, we included only Wait 2 in our analysis because data for Wait 2 were complete; analysis of Wait 1 showed broadly similar results. Some patients, particularly those with more complex disease or atypical presentation, would have a diagnostic phase between initial surgical consultation and the decision to proceed with surgery. Similarly, our analysis does not account for the time from initial presentation or onset of symptoms and referral for surgical consultation. Finally, our analysis did not explicitly account for geographic variation in wait time. A coordinated referral system would need to account for regional variations and referral pathways to limit externalizing the cost of accessing care to patients.

## Conclusion

There is striking variability in surgical wait times for five common, non-urgent surgeries in Ontario. This suggests a lack of coordination where each patient is part of a disconnected queue linked to a single surgeon. Our findings highlight allocative inefficiency, necessitating a re-evaluation of the supply-demand dynamics.

## Acknowledgments

The authors thank Peter Austin and Jiming Fang for methodological advice. This study was supported by ICES. This document used data adapted from the Statistics Canada Postal Code

Conversion File, which is based on data licensed from Canada Post Corporation. Parts of this material are based on data and/or information compiled and provided by MOH and Ontario Health. The analyses, conclusions, opinions and statements expressed herein are solely those of the authors and do not reflect those of the funding or data sources; no endorsement is intended or should be inferred.

## Author Contributions

**Conceptualization:** Pieter de Jager, Dionne Aleman, Nancy Baxter, Chaim Bell, Merve Bodur, Andrew Calzavara, Robert Campbell, Michael Carter, Scott Emerson, Anna Gagliardi, Jonathan Irish, Danielle Martin, Samantha Lee, Marcy SaxeBraithwaite, Pardis Seyedi, Suting Yang, Claudia Zanchetta, David R. Urbach.

**Data curation:** David R. Urbach.

**Formal analysis:** Andrew Calzavara.

**Funding acquisition:** David R. Urbach.

**Methodology:** Pieter de Jager, Andrew Calzavara, Michael Carter, David R. Urbach.

**Project administration:** Pieter de Jager, Samantha Lee, Julie Takata, David R. Urbach.

**Resources:** David R. Urbach.

**Supervision:** David R. Urbach.

**Validation:** David R. Urbach.

**Writing – original draft:** Pieter de Jager, David R. Urbach.

**Writing – review & editing:** Pieter de Jager, Dionne Aleman, Nancy Baxter, Chaim Bell, Merve Bodur, Andrew Calzavara, Robert Campbell, Michael Carter, Scott Emerson, Anna Gagliardi, Jonathan Irish, Danielle Martin, Samantha Lee, Marcy SaxeBraithwaite, Pardis Seyedi, Julie Takata, Suting Yang, Claudia Zanchetta, David R. Urbach.

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
