## [Decision Letter · Decision Letter 0]

20 May 2024

PONE-D-24-09785Surgeon- and hospital-level variation in wait times for scheduled non-urgent surgery in Ontario, Canada: A cross-sectional population-based studyPLOS ONE

Dear Dr. de Jager,

Thank you for submitting your manuscript to PLOS ONE. After careful consideration, we feel that it has merit but does not fully meet PLOS ONE’s publication criteria as it currently stands. Therefore, we invite you to submit a revised version of the manuscript that addresses the points raised during the review process. Please submit your revised manuscript by Jul 04 2024 11:59PM. If you will need more time than this to complete your revisions, please reply to this message or contact the journal office at plosone@plos.org. Please include the following items when submitting your revised manuscript:A rebuttal letter that responds to each point raised by the academic editor and reviewer(s). You should upload this letter as a separate file labeled 'Response to Reviewers'.A marked-up copy of your manuscript that highlights changes made to the original version. You should upload this as a separate file labeled 'Revised Manuscript with Track Changes'.An unmarked version of your revised paper without tracked changes. You should upload this as a separate file labeled 'Manuscript'.

We look forward to receiving your revised manuscript.

Kind regards,

Gianni Virgili

Academic Editor

PLOS ONE

Journal Requirements:

"This project was funded by the Canadian Institutes of Health Research through a project grant (no. PJT-166108)."

5. Please ensure that you include a title page within your main document. You should list all authors and all affiliations as per our author instructions and clearly indicate the corresponding author.

**Additional Editor Comments:**

This is an interesting study trying to measure waiting list time variation at the hospital and surgeon level in Ontario.

One reviewer has concerns on statistical analyses. The normalizing transformation of the response variable (wait 2) is reported to normalize the residuals; in fact, the assumption in linear regression is homoskedasticity (similar residuals across covariate levels), rather than the normality of the response variable.

I suggest the authors are clearer regarding what covariates were planned in the protocol as the fixed part of the model, with changes to the protocol explained. Covariates could also be used in exploratory analyses of how the VPC/ICC is reduced once important covariates are introduced one at a time, e.g. age, sex, deprivation index, fragility indexes if available, etc. The authors could also consider whether surgical volume could be introduced as a random slope at the surgeon level: this would allow to test whether low vs high volume surgeons have different wait 2 time.

Reviewers' comments:

Reviewer's Responses to Questions

**Comments to the Author**

1. Is the manuscript technically sound, and do the data support the conclusions?

Reviewer #1: Yes

Reviewer #2: Partly

2. Has the statistical analysis been performed appropriately and rigorously? 

Reviewer #1: Yes

Reviewer #2: I Don't Know

3. Have the authors made all data underlying the findings in their manuscript fully available?

Reviewer #1: Yes

Reviewer #2: No

4. Is the manuscript presented in an intelligible fashion and written in standard English?

Reviewer #1: Yes

Reviewer #2: Yes

5. Review Comments to the Author

Reviewer #1: The paper aimed to quantify the observed variation in wait time due to surgeons and hospitals. Overall, the paper is well-written and structured and used adequate statistical approach to estimate the variance partition coefficient (VPC) for the surgeon and the hospital effects. There are some concerns that could improve the paper if addressed:

1. The Wald test was applied to test the hypothesis H0: VPC=0 vs H1: VPC>0. As the VPC only nonnegative variances are allowed, there is a boundary (i.e., 0) in the variances’ parameter space, and regular inference statistical procedures for such a parameter could be problematic.

2. Although the authors mentioned that fixed effects were not considered, it would be interesting to conduct an analysis incorporating fixed effects, provided such data are available. Are there any independent factors (surgeons/hospitals) associated with a longer time to surgery?

3. What was the significance level value?

4. Which statistical software was utilized in the data analysis?

Reviewer #2: The authors provide a well-written manuscript on surgeon- and hospital-level variation in the time from decision-to-treat until surgery for the top non-urgent procedures in Ontario, Canada. This is a very important topic, but the discussion can be markedly improved. I do encourage a statistical review for the points I mention below.

- Page 4: The sentence “Ontario's benchmark for Wait 2 for the least urgent category of most non-cancer conditions is within 184 days of deciding to proceed with surgery” is helpful. I was going to ask for the other benchmarks as well, but getting to the Outcomes/Exposures section those seem out of scope. So perhaps move this sentence to the latter section. Please also confirm that this are the current benchmark and whether the benchmark has changed over the course of the study period.

- In the section on data sources, please provide additional information on the WTIS. What kind of coverage does the WTIS have? All patients? All procedures? All hospitals (e.g. this excludes specialty hospitals like Shouldice, but I’m not sure about others). This may be particularly important for cataract surgeries if done at private facilities.

- After excluding procedures occurring within 1 year of a previous surgery of the same type, is this expected to remove the second stage of a staged procedure? Please make that explicit

- Please reference Table 1 in the methods section where the exclusions are described. What is the difference between a wait-2 of 0 days (excluded from the study) versus 1 day (included in the study)? Does it not still suggest data validity questions for non-urgent cases?

- I’m going to recommend this study be reviewed by a statistician. My concerns relate to the transformation of the outcome to make the distribution normal. Issues with this is that back-transformation is not always straight forward (smearing factors have been shown to have their own problems), and GLMs with different link functions and distributional families may be well-suited for skewed data, even with zeros. By transforming the outcome, the variability in the outcome is affected (reduced), so if you’re measuring variability, then you’ve effectively reduced the variability in variability, affecting the statistical analysis.

- Please elaborate on the sentence “If every surgeon had the same mean wait time, the VPC for the surgeon estimate would be 0”. As I understood from the methods, this is a measure of variability, not central tendency. Physicians can have the same mean wait-time, but vastly different distributions in wait-times. I think this is partly explained by the next sentence, but some clarity would be helpful.

- Where was the study data analyzed? E.g. ICES vs OH vs WC

- Figure 2 x-axis: modify the labels to every 10 or so to make it readable.

- The paragraph in the discussion beginning with “Our analyses demonstrate that there is no such thing as a “wait list" for elective surgery” can improve with additional context. It makes it sound random; that a patients' wait-time depends on the surgeon and hospital. Rather, I don’t believe it’s random. Where the patient lives (e.g. urban vs rural; travel distance to nearest center; population density), who they (or their referring GP) knows, and competing demands on surgeon and hospital time matter, and are not random.

- The following paragraph states “our findings suggest there are efficiency gains to be made by improving coordination between the supply and demand of services”, but aside from demonstrating variation in wait-times by surgeon and hospital, this statement is not supported by the data.

- The statement “this analysis suggests that reporting mean or median wait times, or the proportion of patients not meeting waiting time targets, is an inadequate measure of system performance in access to care” is also not clearly evident from the results. Why are wait-times inadequate measures of system performance? No data are shown on variability in the proportion of patients receiving surgery within wait-time targets. There may indeed be very little variability on this measure (and therefore distributional efficiency). Does Ontario Health target wait-times or wait-time targets?

- I agree that the especially long-waiters attract the most attention. So I wonder if those >99th percentile are truly outliers (data entry errors) or truly long-waiters that should have been retained in the analysis. Thoughts on this? With the flexibility of GLMs, transformation of the outcome may not be required, so this reason for omission may be moot.

- Again, the analysis did not suggest this: “However, our analysis suggests that increasing the supply of surgical services will not reduce the longest wait times without a coordinated system that better matches demand to supply.” The statement seems to imply that surgeons are standing around idle, when we know that’s not the case. It's an interesting discussion point, but it's not data-driven.

- Is a “Single-entry referral model” equivalent to a centralized referral system? This is an important paragraph. Some additional context about the existing referral system can help here. Are there any incentives to existing referral practices that may pose a barrier to a single-entry referral model? Do you think that busier GPs spend less time trying to get an earlier referral to a specialist? But all this discussion seems to hover around Wait 1. Once the decision to treat is made, what does a single-entry referral model offer? I presume the biggest barrier is OR time (and staff)

- The conclusion mentions inequity, but there was no data on measures of inequity. The point about single-entry referral system is not a conclusion of the study, but a point for discussion and should be removed from the conclusion.

- What software was used for analysis?

6. PLOS authors have the option to publish the peer review history of their article (what does this mean?). If published, this will include your full peer review and any attached files.

Reviewer #1: No

Reviewer #2: No

---

## [Author Response · Author response to Decision Letter 0]

24 Jun 2024

Dr. Gianni Virgili

Academic Editor

PLOS ONE

19 June 2024

Dear Dr. Gianni Virgili 

Thank you for reviewing our manuscript, “Surgeon- and hospital-level variation in wait times for scheduled non-urgent surgery in Ontario, Canada: A cross-sectional population-based study” (REF: PONE-D-24-09785) for publication in PLOS ONE.

We value the reviewer’s considered and thoughtful comments. Included in this resubmission is a:

- Revised manuscript with all changes highlighted in yellow

- Revised manuscript without changes highlighted

- Response to each of the reviewer comments

We look forward to your response.

Sincerely,

Pieter de Jager

---

## [Editor Report · Decision Letter 1]

12 Jul 2024

Surgeon- and hospital-level variation in wait times for scheduled non-urgent surgery in Ontario, Canada: A cross-sectional population-based study

PONE-D-24-09785R1

Dear Dr. de Jager,

We’re pleased to inform you that your manuscript has been judged scientifically suitable for publication and will be formally accepted for publication once it meets all outstanding technical requirements.

Kind regards,

Gianni Virgili

Academic Editor

PLOS ONE
---

## [Editor Report · Acceptance letter]

16 Jul 2024

PONE-D-24-09785R1 

PLOS ONE

Dear Dr. de Jager, 

I'm pleased to inform you that your manuscript has been deemed suitable for publication in PLOS ONE. Congratulations! Your manuscript is now being handed over to our production team.

Kind regards, 

on behalf of

Dr. Gianni Virgili 

Academic Editor

PLOS ONE